# Dimensions of Alexithymia and Identification of Emotions in Masked and Unmasked Faces

**DOI:** 10.3390/bs14080692

**Published:** 2024-08-09

**Authors:** Thomas Suslow, Anette Kersting, Charlott Maria Bodenschatz

**Affiliations:** Department of Psychosomatic Medicine and Psychotherapy, University of Leipzig Medical Center, 04103 Leipzig, Germany; anette.kersting@medizin.uni-leipzig.de (A.K.); charlott.bodenschatz@medizin.uni-leipzig.de (C.M.B.)

**Keywords:** externally oriented thinking, difficulties describing feelings, difficulties identifying feelings, alexithymia, emotion recognition, facial emotions, face masks

## Abstract

Alexithymia, a multifaceted personality construct, is known to be related to difficulties in the decoding of emotional facial expressions, especially in case of suboptimal stimuli. The present study investigated whether and which facets of alexithymia are related to impairments in the recognition of emotions in faces with face masks. Accuracy and speed of emotion recognition were examined in a block of faces with and a block of faces without face masks in a sample of 102 healthy individuals. The order of blocks varied between participants. Emotions were recognized better and faster in unmasked than in masked faces. Recognition performance was worst and slowest for participants starting the task with masked faces. In the whole sample, there were no correlations of alexithymia facets with accuracy and speed of emotion recognition for masked and unmasked faces. In participants starting the task with masked faces, the facet externally oriented thinking was positively correlated with reaction latencies of correct responses for masked faces. Our findings indicate that an externally oriented thinking style could be linked to a less efficient identification of emotions from faces wearing masks when task difficulty is high and support the utility of a facet approach in alexithymia research.

## 1. Introduction 

The accurate identification of emotions in other people’s faces is crucial for social life, since it promotes smooth social interactions and helps to build social relations and bonds [1,2,3]. The perception of facial emotions is complex and appears to involve a number of different processes and brain systems [4,5]. It is assumed that the identification of emotional facial expressions could be based on the processing of single or multiple features of a face, which are diagnostically relevant for an emotion [6,7]. There is also evidence that holistic processing is used for the recognition of emotional facial expressions [8]. It has also been argued that the ability to recognize facially expressed emotions of other individuals relies, at least in part, on processes which simulate the same emotional state in the observer [9]. This means, when people perceive an emotional facial expression, they partially activate the respective emotion in themselves, providing a basis for the recognition of that emotion [10]. In general, accuracy of facial emotion recognition is rather high for most of the basic emotions [11,12] and linked to their frequency in everyday life [13]. The relevance of recognizing emotions in others’ faces makes the identification of personality factors influencing facial emotion recognition highly important.

During the COVID-19 pandemic, social life has been shaped by containment measures such as mask wearing and social distancing. Face masks reduce viral transmission and have been an important component in preventing the spread of COVID-19 [14]. However, the wearing of face masks has adverse effects on the capacity to read facial expressions, making it more difficult to recognize quality and intensity of other people’s emotions [15,16]. In recent years, substantial evidence has accumulated that face masks hamper the recognition of basic emotions such as happiness, sadness, disgust, and fear [17,18,19,20].

Alexithymia is a multifaceted personality trait primarily defined by difficulties in identifying and verbalizing one’s feelings and an externally oriented thinking style [21]. It has been shown that individuals with alexithymic characteristics do not have only difficulties in identifying their own emotions but also those of other people. Alexithymia has been found to be linked to impairments in recognizing others’ emotional facial expressions even when these expressions are very intense and displayed for a long duration [22,23]—but see Pandey and Mandal [24] and McDonald and Prkachin [25] for null results. Findings concerning alexithymia-related difficulties in perceiving emotions in faces are more consistent in research with suboptimal presentation of facial stimuli, i.e., showing faces with temporal constraints (for example for only 66 or 100 ms) [26,27,28] or in degraded quality (as blurred images) [29]. According to the systematic review of Grynberg et al. [30] alexithymic individuals’ difficulties in decoding facial emotions are neither limited to a specific emotional valence nor a specific emotional quality. That is, alexithymia appears to be associated, for example, with impairments in identifying facial happiness as well as with difficulties in recognizing facial sadness, fear, anger, or disgust.

In a recent integrative review of the literature on cognitive–emotional processing in alexithymia, Luminet et al. [31] underline the importance of considering the alexithymia facets separately. There is evidence for differential relationships of the alexithymia facets with cognitive–emotional functioning across various domains (attention, appraisal, memory, language, and behavior). The influence of the externally oriented thinking facet seems rather distinct from the influence of difficulties in identifying and describing feelings. Externally oriented thinking appears to be a central facet involved in emotion processing impairments, in particular, concerning attentional and appraisal processes [31,32] (see also [27,29,33,34]).

Three previous studies based on samples of non-clinical individuals have so far investigated the effect of alexithymia on emotion identification in faces with face masks. Fuchs et al. [35] compared highly alexithymic with non-alexithymic individuals in an emotion recognition task in which masked faces expressing happiness, fear, and disgust were presented. Study participants were classified as alexithymic or non-alexithymic using the cut-offs provided by Bagby and Taylor [36]. In the former study [35], no group differences were observed for accuracy and speed of emotion recognition in masked faces. Verroca et al. [37] focused their dimensional analysis on recognition accuracy and examined alexithymia as a unitary construct, i.e., using the total alexithymia score. In their recognition task, which was conducted online, participants had to identify six basic emotions (i.e., happiness, surprise, anger, disgust, fear, and sadness) in blocks of unmasked and masked faces with a randomized order of blocks across participants. Verroca et al. [37] found no association between overall alexithymia and accuracy in identifying emotions from masked faces. The only investigation that followed a facet approach of alexithymia was conducted by Maiorana et al. [38]. In this study, masked faces with expressions of happiness, anger, and sadness were presented as stimuli. The authors found no correlations between alexithymia facets and speed of emotion identification in faces with masks. The null findings in this study might be due to a rather small sample size (*N* = 31). In addition, it can be argued that the authors presented different types of emotional facial stimuli to their participants (unmasked faces, masked faces, lower face parts, upper face parts) in a counterbalanced order (in a block format) but did not consider the effect of order on the relation between alexithymia and emotion recognition in masked faces.

The main objective of the present study was to explore the recognition of emotions in faces with a face mask as a function of alexithymia facets. In our study, we administered a block of faces with face masks and as a control condition a block of faces without face masks. It was expected that face masks have a negative impact on the recognition of facial emotions. In the present study, we varied the order of blocks between participants. First, as in the previous dimensional alexithymia studies [37,38], we analyzed emotion recognition in the whole sample regardless of the order of blocks in which masked and unmasked faces were presented. Second, we conducted separate correlation analyses for individuals who started the recognition task with masked faces and those who started with unmasked faces.

It can be assumed that when participants are first confronted with the identification of emotions in masked faces the task would be more difficult for them compared to the condition in which participants are first confronted with the identification of emotions in unmasked faces. In the latter case, participants become familiar with the task-relevant emotion qualities and the response format. It can be hypothesized that the identification of masked facial emotions is most difficult when the task starts with masked faces. As the duration of facial stimulus presentation was rather long in our study (with a maximum of 5 s) we expected that alexithymia would be related to a slowing of response latencies. The association between alexithymia and response latency should be most pronounced for the most difficult task condition, i.e., masked faces presented as the first experimental block. In our emotion recognition task, five different categories of facial expressions were shown, i.e., happy, sad, fearful, disgusted, and neutral faces.

Following the recommendations of a recent review [31] we administered a dimensional facet approach to examine processes of facial emotion recognition in alexithymia. Results from previous research indicate that the facet externally oriented thinking is primarily involved in emotion processing impairments [27,29,31]. Against this background, we assumed that especially externally oriented thinking is related to difficulties in facial emotion recognition. We assessed and controlled participants’ dispositional anxiety and level of depressive symptoms since these variables have been found to be associated with alexithymia [39,40]. Participants’ intelligence was measured in our study as it can influence decoding of facial emotions [41]. Since we were interested in the speed of emotion recognition, which involves motor responses by pressing buttons on a keyboard, the Trail Making Test Part B [42] was used to assess visuomotor processing speed in a task with non-emotional stimuli.

## 2. Materials and Methods

### 2.1. Participants

One hundred and five young volunteers participated in the present study. The final sample consisted of one hundred and two participants (seventy women) with a mean age of 24.05 years (SD: 4.25) and a mean school education of 12.21 years (SD: 1.01). Three participants were removed from the analyses because of slow response times in the emotion recognition task (see for details Section 2.5 below). Participants were recruited via public notices and online advertisements. All participants were native German speakers and had normal vision as tested with a Snellen eye chart. According to self-report they were free of any lifetime history of psychiatric or neurological disorders and did not use psychotropic medication. About three quarters of the participants were university students from various disciplines. The study protocol was approved by the local ethics committee. Before the experiment, informed written consent was obtained from each participant.

### 2.2. Questionnaires and Tests

The 20-Item Toronto Alexithymia Scale (TAS-20) is a self-report measure that assesses three facets of alexithymia [43,44]: difficulty identifying feelings (DIF), difficulty describing feelings (DDF), and externally oriented thinking (EOT). Items are rated on a 5-point scale. The total alexithymia score is the sum of responses to all 20 items with a range of possible scores from 20 to 100. Higher TAS-20 scores indicate higher alexithymia. In the present sample, Cronbach’s alphas for the subscales DIF, DDF, and EOT and the total score were 0.75, 0.85, 0.64, and 0.85 which were similar to the coefficients reported by Bagby et al. [43] for a sample of university students: 0.79, 0.75, 0.66, and 0.80.

The Beck Depression Inventory (BDI-II) is a 21-item, self-rated scale that assesses characteristic attitudes and symptoms of depression during the preceding two weeks [45,46]. Each item is rated on a 4-point scale ranging from 0 to 3. The maximum total score of the BDI-II is 63. Cronbach’s alpha was 0.77 for the BDI-II in our sample.

The State-Trait Anxiety Inventory (STAI) trait version was administered to assess participants’ trait anxiety [47,48]. Trait anxiety refers to a general tendency to respond with anxiety to perceived threats and is a relatively stable personality characteristic. The STAI contains 20 items that are scored on a 4-point scale (from 1 to 4). Total scores range between 20 and 80. In our study, Cronbach’s alpha was 0.89 for the STAI.

The Trail Making Test Part B (TMT-B) assesses executive functioning, visual attention, and visuomotor processing speed [42]. In this paper-and-pencil test, participants have to connect numbers and letters in ascending order (twenty-five items in all). The TMT is scored by how long it takes to complete the test (in seconds). Higher scores indicate slower switching between numbers and letters.

The Mehrfachwahl-Wortschatz-Intelligenztest Version B (MWT-B) is a multiple-choice vocabulary intelligence test [49] that measures aspects of general intelligence, specifically crystallized, verbal intelligence. The MWT-B consists of 37 items. Each item comprises one real word and four pronounceable pseudo-words. The subject has the task to identify and cross out the real word. Sum scores of correct responses are calculated and can be converted to IQ scores.

### 2.3. Emotion Recognition Task: Stimuli and Procedure

Facial stimuli comprised 100 color photographs of twenty young models (ten women) from the MPI FACES database [50]. Each model was shown with five different facial expressions (happy, sad, fearful, disgusted, and neutral). Ten models (five women) were presented without a mask as original FACES images. The photos of ten other models (five women) were digitally adapted by superimposing a light blue mask on the original faces. The masks were adapted to match the width and length of the faces so that they covered faces from the upper nose downwards. The display size of each face photo on the screen was 10.2 cm high × 8 cm wide.

The emotion recognition task consisted of 100 trials divided into two experimental blocks. In one block, faces without a mask were shown (50 trials) and, in another block, faces with a mask were displayed (50 trials). About half of the sample saw first the block with the original (unmasked) faces and then the block with the faces wearing masks (n = 54, “start with unmasked”). The other half of the sample saw the blocks in reverse order (n = 51, “start with masked”). At the beginning of the task, participants were instructed that they would see faces on the screen expressing fear, disgust, sadness, or happiness. They were also informed that some faces in the task have neutral expressions. Participants were told to identify the expressed emotions and to respond primarily as accurately but also as quickly as possible. We registered responses and reaction latencies of participants. Experimental trials in each block were presented in an individual random order.

Each trial in the emotion recognition task had the same structure: after the appearance of a central fixation cross, which was shown for 1000 ms, a face stimulus was displayed for a maximum of 5000 ms. Participants responded by button press in a forced choice manner. Responses were given on a keyboard using the number keys 1 (happiness), 2 (sadness), 3 (neutral), 4 (disgust), 5 (fear). Each emotion was assigned to one key. The expression categories and the assigned numbers were shown at the bottom of the screen in black letters during the entire experiment.

During the task participants were seated in a chair at approximately 60 cm in front of the screen. The computer-based stimulus presentation and response registration were realized via PsychoPy 3 version 2020.2.4 [51] on a Dell Latitude E6540 with a 15.6-inch screen.

### 2.4. General Procedure

At the beginning of the experiment, participants completed a sociodemographic questionnaire and performed a vision test (Snellen eye chart). Subsequently, participants performed the computer-based emotion recognition task. At the end of the session, the questionnaires and tests were administered in a fixed order: BDI-II, STAI trait, MWT-B, TAS-20, and TMT-B. All testing sessions were conducted individually in a quiet laboratory room.

### 2.5. Statistical Analyses

The dependent variables in the emotion recognition task were accuracy and response time. Accuracy was scored as the number of correctly identified emotional expressions. Response times were calculated for all correct trials across expression conditions, resulting in an overall response latency score for masked and unmasked faces. In addition, we calculated the mean number of hits and the mean response latency for correct responses for each expression condition. The 105 participants of our study gave 10,500 responses in the emotion recognition task. Of them, 9694 responses were correct (92.3%). Response latencies were in no trial below 500 ms. Thus, there was no evidence for unrealistically fast decisions. We excluded very slow responses or responders from data analysis. Thirteen response latencies were above 10 s and were removed from further data processing. To identify slow responders, we analyzed the resulting overall response latencies of all participants. A criterion of three standard deviations from the mean was used to find outliers. The upper cut-off score was 3278 ms for the overall response latency for unmasked faces and 3606 ms for the overall response latency for masked faces. In this way, three participants were identified as slow responders (two women, one man) and excluded from data analysis so that our final sample consisted of one hundred and two individuals. In our study, no participant exhibited overall response latency scores, which were three standard deviations below the means.

Kolmogorov–Smirnov tests were applied to assess normality of distribution. There was a significant departure from normality for the majority of the investigated variables (i.e., DIF, DDF, EOT, BDI-II, STAI-T, MWT-B, TMT-B, overall hits (unmasked faces), overall hits (masked faces), RT (masked faces), all *p*s < 0.05). Against this background, we administered non-parametric tests to examine the effect of face masks on accuracy and speed of correct facial emotion recognition (Wilcoxon tests for paired samples). To examine the effect of order of presentation on accuracy and speed of emotion recognition we used Mann–Whitney U tests comparing the participants who started the task with masked faces with the participants who started with unmasked faces. Mann–Whitney U tests were also used to compare these two participants groups on the questionnaire (TAS-20, BDI-II, and STAI-T) and test variables (TMT-B and MWT-B). Spearman rank correlation analysis was performed to examine the relationships between TAS-20, measures of negative affect, visuomotor processing speed, intelligence, and performance in the emotion recognition task. All calculations were made with SPSS 29.0 (IBM Corp., Armonk, NY, USA). Results were considered significant at *p* < 0.05, two-tailed. To adjust for multiple testing in our main correlation analyses concerning the associations of alexithymia facets with emotion recognition parameters, we used a conservative significance threshold of 0.05/24 = 0.00208 (i.e., dividing the conventional p-level by the product of the two masking conditions (masked, unmasked), the two block sequences (start with masked, start with unmasked), the two recognition parameters (accuracy, latency), and the three alexithymia facets (DIF, DDF, and EOT)). In these correlation analyses, one-tailed testing was applied because the hypotheses had a direction.

We calculated an (a priori) power analysis with the program G*Power 3.1 [52] to determine the required sample size to detect correlations between the alexithymia facet externally oriented thinking and emotion recognition in masked faces. We based our analysis on the large effect size of 0.58 observed in healthy individuals for the relation between externally oriented thinking and recognition of emotional facial expressions in degraded faces (see Kätsyri et al. [29], Table 2). To detect a medium effect of *r* = 0.58 with an alpha value of 0.05, two-tailed, and a statistical power of 0.95, the required sample size is 28 (type of power analysis: a priori—compute required sample size given α, power, and effect size). Thus, the number of participants in the subgroups of our study should be at least 28.

## 3. Results

### 3.1. Emotion Recognition Performance

Participants’ recognition performance (number of correct responses or hits) is presented in Figure 1 as a function of masks and order of blocks. We analyzed in the whole sample whether number of correct responses differed between the masked and the unmasked face condition. According to the results of a Wilcoxon test, the number of correctly identified facial emotions was significantly higher in the unmasked (mean: 9.55, SD: 0.39) compared to the masked face condition (mean: 8.91, SD: 0.48), *Z* = −7.85, *p* < 0.001.

According to the results of a Mann–Whitney U test, participants starting with masked faces showed no differences in number of hits for unmasked faces compared to participants starting with unmasked faces, *Z* = −1.31, *p* = 0.19. In contrast, participants starting with masked faces had a lower number of hits for masked faces compared to those starting with unmasked faces, *Z* = −2.86, *p* < 0.005.

Response times for correct responses in the emotion recognition task are shown in Figure 2 as a function of masks and order of blocks. In the whole sample, response latencies of correct responses differed between the masked and the unmasked face condition. Response times were significantly lower in the unmasked (mean: 1.96 s, SD: 0.35) compared to the masked face condition (mean: 2.15 s, SD: 0.39), *Z* = −4.11, *p* < 0.001.

Participants starting with unmasked faces were slower in identifying emotions in unmasked faces compared to participants starting with masked faces, *Z* = −5.51, *p* < 0.001. Moreover, participants who began the recognition task with unmasked faces were faster to identify emotions in masked faces compared to those who started with masked faces, *Z* = −3.01, *p* < 0.005. Interestingly, results from Wilcoxon tests indicated that response times were faster in the second block irrespective of masking type encountered in the first block. In individuals starting with unmasked faces response latencies were higher for unmasked faces than for masked faces, *Z* = −2.76, *p* < 0.01, whereas in individuals starting with masked faces response latencies were higher for masked faces than for unmasked faces, *Z* = −6.09, *p* < 0.001 (see Figure 2).

### 3.2. Correlations of Alexithymia with Psychological Tests

The correlation analysis indicated that the sum score and the subscales DIF and DDF of the TAS-20 were positively related to depressed symptoms as assessed by the BDI-II (see Table 1 for details). Moreover, the sum score and all subscales of the TAS-20 were positively correlated with trait anxiety (STAI-T). The sum score and the subscales DIF and DDF of the TAS-20 showed negative correlations with the TMT-B. Finally, no significant correlations were observed between TAS-20 scales and intelligence (MWT-B) (see Table 1).

### 3.3. Correlations of Alexithymia with Recognition Performance

The results of Spearman rank correlations between TAS-20 scales and emotion recognition are shown in Table 2. In the total sample, when administering the adjusted p-level of 0.00208 we found no significant correlations of the TAS-20 sum score and the scales DIF, DDF, and EOT with the overall number of correct responses and the overall response latencies for the masked and unmasked expression conditions.

### 3.4. Correlations of Affectivity, Psychomotor Functioning, and Intelligence with Recognition Performance

No correlations were found between trait anxiety, psychomotor functioning, intelligence, and the overall number of hits and the overall response latencies for unmasked and masked faces. Participants’ level of depressive symptoms as assessed by the BDI-II was positively correlated with RT for correct responses in the unmasked condition but not with RT in the masked condition (see Table 3).

### 3.5. Comparisons between Participants Starting with Unmasked Faces and Participants Starting with Masked Faces Concerning Alexithymia, Affectivity, Psychomotor Functioning, and Intelligence

According to the results of Mann–Whitney U tests, participants starting with unmasked faces in the emotion recognition task (n = 52; 34 women) did not differ from participants starting with masked faces (n = 50; 36 women) on the TAS-20 scales (see Table 4). Moreover, no group differences were observed for level of depressive symptoms, trait anxiety, and intelligence. Participants starting with unmasked faces manifested slower psychomotor functioning compared to those starting with masked faces (see Table 4).

### 3.6. Correlations of Alexithymia with Recognition Performance in Participants Starting with Unmasked Faces and Participants Starting with Masked Faces

In the sample of participants who started the emotion recognition task with unmasked faces, there were no significant correlations of the TAS-20 scales with the overall number of correct responses and the overall response latencies for the unmasked and masked expression conditions (see Table 5). In the sample of participants starting with the masked faces, there were also no correlations of the TAS-20 scales with the overall number of correct responses for the unmasked and masked expression conditions. However, EOT showed a significant positive correlation with overall response latencies for the masked expression condition (see Table 5).

Additional correlation analyses were carried out in the sample of participants who started with masked faces to explore the relationships between EOT and response latencies for the five masked emotional expression conditions (see Table 6). There was only a significant positive correlation of EOT with response time for masked disgusted faces. No other significant correlations were observed.

To compare response latencies between masked facial expression conditions in participants starting with masked faces we conducted Wilcoxon tests. The results indicated that response time for correctly identified masked disgust was significantly longer than the response times for correctly identified masked neutral, happy, fearful, and sad expressions (all *p*s < 0.05). Moreover, response time for correctly identified masked sadness was significantly longer than the response times for correctly identified masked neutral and happy facial expressions (*p*s < 0.001) but did not differ from response time for masked fear. This means that when considering response latency of correct responses masked disgust was the most difficult emotion to identify followed by masked sadness and masked fear.

Finally, we compared hit rates between masked facial expression conditions in participants starting with masked faces using Wilcoxon tests. Hit rate for masked sad faces (6.88 (SD = 1.42)) was significantly lower than hit rates for masked neutral (9.94, SD = 0.24), fearful (9.30, SD = 1.33), disgusted (8.84, SD = 1.48), and happy faces (8.82, SD = 0.66) (all *p*s < 0.001). Hit rate for masked disgust did not differ from hit rates for masked happy and fearful facial expressions. Thus, when considering the number of correct responses masked sadness was the most difficult emotion to identify in the emotion recognition task.

## 4. Discussion

The correct identification of emotions in facial expressions is important for successful interaction with other people [1,2,3]. Since the start of the COVID-19 pandemic, face masks were used to prevent the spreading of the coronavirus and became part of public life. Unfortunately, the occlusion of the mouth and nose has an adverse impact on facial emotion recognition. The identification of basic emotions in other people’s facial expressions is substantially worse for masked compared to unmasked faces though it leaves inferring basic emotional expressions clearly above chance level [18,19]. The covering of the lower face hides facial features relevant for emotion recognition [7,53] and it could also disrupt holistic face processing [16]. The data of our study clearly confirm that emotions are recognized better and faster in unmasked faces than in masked faces. Moreover, we observed that recognition performance was worst and slowest for participants who started the task with masked faces.

In the current study, we examined whether alexithymia dimensions are associated with difficulties in the recognition of emotions in faces with a face mask. We focused in our analyses on alexithymia facets since results from previous research on cognitive–emotional processing in alexithymia highlighted the utility of a facet approach [31,32]. In the whole sample, we found no correlation of alexithymia facets with accuracy or speed of emotion recognition for masked and unmasked faces in the emotion recognition task. Our null findings concerning recognition speed in the whole sample correspond to those observed in the only previous investigation, which followed a facet approach to study the relations between alexithymia and emotion decoding in faces wearing masks [38]. In this study, no correlations were detected between alexithymia facets and speed of emotion identification in masked faces. Interestingly, Maiorana et al. [38] presented different types of emotional facial stimuli, i.e., unmasked faces, masked faces, lower face parts, and upper face parts, in a counterbalanced block format. However, they pooled the recognition data for the conditions across block positions and did not analyze the effect of order on recognition performance. It is likely that participants who began the task with emotion recognition in masked faces responded more slowly compared to those who encountered the masked face stimuli at the end of the recognition task. The aggregation of data across block positions could have made it more difficult to reveal associations between alexithymia facets and speed of emotion decoding. Future research on alexithymia and emotion perception should pay more attention to the difficulty and variability of difficulty in emotion recognition tasks with multiple stimulus conditions, in particular, when samples with rather low levels of alexithymic characteristics are examined.

Consistent with our hypothesis, we found externally oriented thinking to be linked to high response latencies for correct identifications of emotions in masked faces. This association was revealed in participants who started the task with masked faces, i.e., the most difficult task condition. This correlation was of medium to large effect size. Thus, the alexithymia facet externally oriented thinking seems to be linked to slower emotion recognition in faces wearing masks when participants are unprepared, i.e., when they are not familiar with the task. This relationship did not emerge for participants who started the task with the unmasked faces. It is conceivable that being familiar with the task requirements may attenuate impairments in emotion perception linked to externally oriented thinking. According to the results of additional analyses externally oriented thinking was found to be related to long response times in the recognition of masked disgust. Thus, high levels of externally oriented thinking could be associated with slowed recognition of masked disgusted expressions. However, it seems advisable to interpret this finding cautiously in terms of emotion specificity or valence. In our experiment, masked disgust was the most difficult emotional quality to detect when considering latencies of correct responses. Against this background it is possible that the correlations of externally oriented thinking with decoding of disgust in masked faces are due to their general recognition difficulty.

Importantly, in our study individuals starting the recognition task with masked faces did not differ from those starting with unmasked faces with regard to alexithymic characteristics, level of depressive symptoms, trait anxiety, and intelligence. A group difference was revealed for psychomotor functioning. However, psychomotor functioning was not correlated with response latencies in the masked (and unmasked) expression conditions of our emotion recognition task.

Overall, the present data suggest that externally oriented thinking could be associated with less efficient emotion processing in faces wearing masks. The covering of the lower face by face masks reduces the available information on the expressed emotions. For individuals high in externally oriented thinking such a shortage of available information seems to cause problems in rapidly identifying the correct emotional expression. This alexithymia facet could be characterized by less efficient reading out and use of perceptual and lexical emotional information in the labeling of facial expressions. Our results are in line with and confirm the conclusions of recent reviews in the research field that in particular externally oriented thinking is crucially connected with impairments in attentional and appraisal processes during emotion perception [27,31,32,33]. The observed rather high correlation between externally oriented thinking and emotion recognition performance in our study confirms the relation reported by Kätsyri et al. [29], who presented blurred faces briefly (with a duration varying from 0.8 to 2 s). Interestingly, in both studies facial stimuli were characterized by a removal of information on facial features which are assumed to be relevant for emotion recognition. In the study of Kätsyri et al. the removal of information by image degradation with a loss of details concerned the surface of the whole face, whereas in our study the lower half of the face was completely covered and not available for processes of emotion decoding. A substantial reduction in the amount of available visual information on facial features may lead, in particular in individuals high in externally oriented thinking, to a slowing of emotion recognition processes.

In a previous eye-tracking study [35], we compared identification of emotions in masked faces between highly alexithymic and non-alexithymic individuals. Fuchs et al. [35] found that highly alexithymic participants did not differ from non-alexithymic participants in accuracy and speed of emotion decoding. In the recognition task administered, facial stimuli were also selected from the FACES database and consisted of happy, disgusted, fearful, and neutral expressions. Different from the present study, sadness was not included as a facial expression category. In the study of Fuchs et al. and the present study, the masks administered resembled light blue surgical face masks and covered the faces from the upper nose downwards. The overall hit rate for masked faces in Fuchs et al.’s study (0.87, SD = 0.08) was similar to that observed in the present study (0.89, SD = 0.05). The overall response latency for correct responses for masked faces was substantially higher (3.55 s, SD = 0.46) than in the present study (2.15 s, SD = 0.39). This difference in response latency could be due to different response modes and/or differences in stimulus presentation time: in Fuchs et al.’s study, expression categories were shown on the screen for only 2 s and participants made their decisions by mouse clicking on response fields, while in the present study facial stimuli were presented for a maximum of 5 s and participants gave their answers by pressing number keys on a keyboard. The latter response mode and longer duration of stimulus presentation might have accelerated responding in the present study. In both studies, participants were young and healthy adult individuals. The mean age of participants in Fuchs et al.’s study (24.37, SD = 4.67) was similar to that in the present study (24.05, SD = 4.25). There was no overlap of samples between studies. The most important difference between studies concerned the analytical approach. Fuchs et al. followed a categorical approach of analysis treating alexithymia as a unitary construct and assigning selected participants to extreme groups (defined by presence or absence of alexithymia) based on the TAS-20 total score, whereas in the present study a dimensional facet approach of analysis was chosen. Highly alexithymic individuals can differ considerably from one another in the degree of specific facets and represent a rather heterogeneous group. Some alexithymic persons may be primarily characterized by difficulties in describing their feelings, while others can be primarily characterized by an externally oriented thinking style. The different patterns of results between studies indicate that a dimensional facet approach could be more useful to detect alexithymia-related impairments in emotion decoding than a categorical approach [35]. Recently, research has begun to pay more attention to analyzing the influence of individual facets of alexithymia through subscale scores, because facet scores allow clearer identification of which alexithymia features are particularly related to specific emotion processing impairments [31,32]. Therefore, it seems advisable to prioritize a facet approach over a categorical or total score approach in studies on alexithymia and emotion perception.

In our study, none of the alexithymia facets was related to recognition accuracy for masked and unmasked faces in the total sample or the subsamples. This finding could be due to the relatively long stimulus presentation time in our experiment, which may have contributed to the high overall percentage of correct responses in our recognition task (95.5% for unmasked faces and 89.1% for masked faces). Study participants had considerable time to make their decisions. It would be interesting to examine in future studies emotion identification under conditions of time pressure by using a response-window technique and short stimulus presentation times. Possibly, under such experimental conditions, emotion processing impairments associated with externally oriented thinking would become manifest in the form of errors. The present data indicate that, not surprisingly, order of presentation can affect task difficulty and especially in samples characterized by low levels of alexithymia it seems advantageous to administer difficult recognition tasks to increase the probability to detect alexithymia-related impairments in emotion decoding.

Emotional facial expressions of other people provide useful clues to their motives, intentions, and attitudes [54]. The ability to read facial expressions is important for social functioning, as it helps to coordinate relationships with other people [55]. Unsurprisingly, deficits in decoding emotions from faces have been found to be linked to interpersonal difficulties and a reduced ability to create and to preserve positive social relationships [56,57]. There is evidence that an externally oriented thinking style goes along with impairments in interpersonal relations. Individuals high in externally oriented thinking have difficulties to empathize and appear self-centered [58,59]. Impaired perception of facial emotions might contribute to dysfunctional interpersonal relationships in alexithymic individuals. When communicating with people wearing face masks interpersonal misunderstanding may occur more frequently in externally oriented individuals and could exacerbate their social problems. Beyond the pandemic, our findings could be of importance for clinicians who use face masks in their everyday professional life. Successful nurse–patient or doctor–patient communications require correct and fast interpretation of emotional states. If clinicians encounter patients characterized by externally oriented thinking and have to communicate with them while wearing face masks, it could be advantageous to intensify or extend the duration of their facial emotions or to make greater use of vocal emotional communication and verbal descriptions of emotions.

The results of our study may point to a positive aspect concerning the modifiability of alexithymia-related difficulties in recognizing emotions from masked faces. Externally oriented thinking was not linked to emotion recognition in masked faces when immediately before emotion identification was “trained” using unmasked facial expressions. However, before firm conclusions can be drawn on this point, there is a need to further investigate the effect of training interventions on emotion recognition abilities in alexithymia. Smartphones represent flexible devices that can play an important role in training facial emotion identification. Recently, Lukas et al. [60] developed a psychological intervention to reduce alexithymia, which combines psychoeducation with a smartphone-based emotion recognition skills training.

A limitation of our study is that alexithymia was only assessed with a self-report measure. Self-descriptive tests assessing alexithymia have been criticized, since they depend on the abilities to describe and attend to one’s feelings correctly [61]. However, over recent decades, research has provided considerable evidence for the reliability and validity of the TAS-20 [62]. Moreover, it can be argued that individuals with higher levels of education could be well aware of their own alexithymic characteristics due to the integration of negative social feedback concerning their difficulties in perceiving, feeling, and communicating emotions regarding their self-concept. It has to be mentioned as limitations that our study participants were healthy, young, and predominantly female individuals and that the mean TAS-20 alexithymia sum score was relatively low in our sample. This limits the generalizability of our results. Range-restriction effects associated with using a relatively homogeneous healthy sample could have reduced the magnitude of relationships observed. Another limitation is that our masked faces represent artificial stimuli, which may not have ecological validity. Our facial stimuli were created by graphically imposing an image of a blue face mask over original photos of emotional facial expressions taken from the MPI FACES database [50]. However, it is worth outlining that recently Grenville and Dwyer [63] compared the accuracy of emotion recognition from graphically created facial stimuli with that from stimuli where the emotional expressions were posed by people who wore face masks. Interestingly, the authors observed quite similar effects on emotion recognition for both types of facial stimuli. It is also a limitation of our investigation that static images of intense facial expressions were administered in our recognition experiment, yet emotional facial expressions in daily life are in general dynamic and frequently more subtle. Finally, it can be criticized that we showed different models in the two masking conditions of our emotion recognition task. To control for differences in expression intensity or purity of expression it would have been methodologically stronger to present the same facial stimuli across masking conditions.

In sum, the present study provides further evidence that occluding lower face parts by face masks has a negative impact on the recognition of basic emotions. The alexithymia facet externally oriented thinking seems to be related to a slowed identification of emotions from faces wearing masks when task difficulty is high. These difficulties in recognizing emotions could lead to less understanding of others’ states, attitudes, and intentions and contribute to the development of problems in interactions with mask-wearing people. For a comprehensive understanding of the impact face masks exert on emotion recognition it seems necessary to consider the personality trait alexithymia. The present findings suggest that a specific facet of alexithymia could be linked to impairments in decoding emotions from faces wearing masks, demonstrating the utility of a facet approach in alexithymia research.

## Figures and Tables

**Figure 1 behavsci-14-00692-f001:**
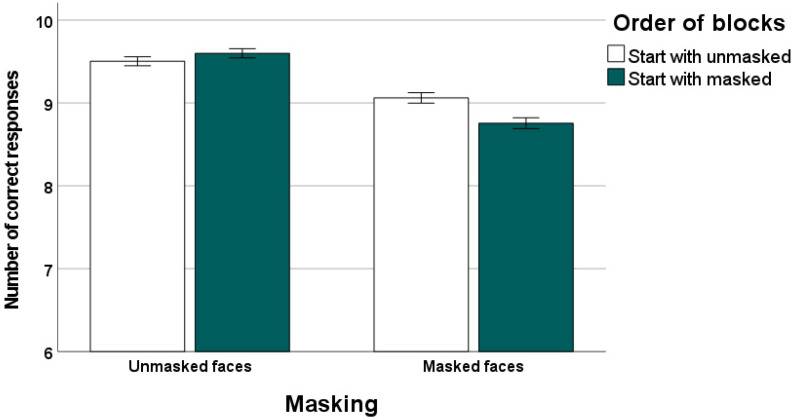
Number of hits (correct responses) in the emotion recognition task as a function of mask and order of blocks, i.e., start with unmasked vs. start with masked faces (error bars: standard error).

**Figure 2 behavsci-14-00692-f002:**
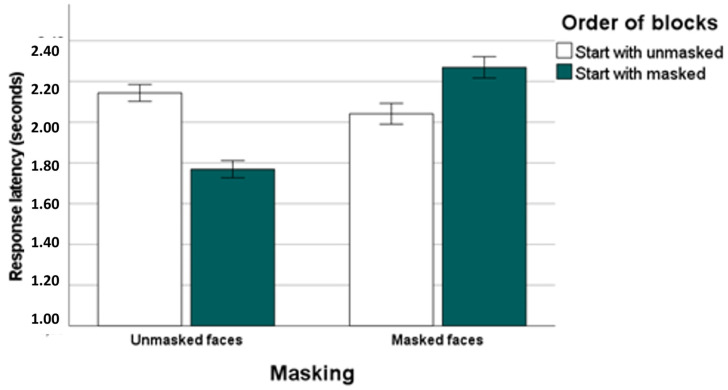
Response times (in seconds) for correct answers in the emotion recognition task as a function of mask and order of blocks, i.e., start with unmasked vs. start with masked faces (error bars: standard error).

**Table 1 behavsci-14-00692-t001:** Correlations between TAS-20 scales and psychological tests (Spearman rank) with descriptive statistics (*N* = 102).

Variable	TAS-20 Sum Score	TAS-20 DIF	TAS-20 DDF	TAS-20 EOT	Mean	SD
TAS-20 (sum score)	---	0.82 ***	0.88 ***	0.67 ***	41.48	10.65
TAS-20 DIF	---	---	0.62 ***	0.29 ***	14.04	4.38
TAS-20 DDF	---	---	---	0.43 ***	11.82	4.59
TAS-20 EOT	---	---	---	---	15.62	4.32
BDI-II	0.29 **	0.34 ***	0.22 *	0.17	8.74	5.13
STAI-T	0.47 **	0.51***	0.36 ***	0.22 *	38.29	8.76
TMT-B	−0.25 *	−0.24 *	−0.21 *	−0.11	56.87	20.01
MWT-B IQ	−0.02	−0.04	−0.01	−0.06	109.27	11.50

* *p* < 0.05; ** *p* < 0.01; *** *p* < 0.001 (two-tailed). TAS-20 = 20-Item Toronto Alexithymia Scale; DIF = Difficulties identifying feelings; DDF = Difficulties describing feelings; EOT = Externally oriented thinking; BDI-II = Beck Depression Inventory; STAI-T = State-Trait Anxiety Inventory—trait version; TMT-B = Trail Making Test Part B; MWT-B = Multiple-Choice Vocabulary Test Version B, intelligence quotient.

**Table 2 behavsci-14-00692-t002:** Correlations of the TAS-20 scales with overall number of correct responses (hits) and overall response latencies for the unmasked and masked expression conditions in the emotion recognition task (Spearman rank) (*N* = 102).

Variable	TAS-20 Sum Score	TAS-20 DIF	TAS-20 DDF	TAS-20 EOT
Unmasked hits	0.03	−0.04	0.01	0.03
Masked hits	−0.05	−0.02	−0.14	0.06
Unmasked RT	0.01	0.00	0.08	−0.02
Masked RT	0.16	0.06	0.21	0.09

TAS-20 = 20-Item Toronto Alexithymia Scale; DIF = Difficulties identifying feelings; DDF = Difficulties describing feelings; EOT = Externally oriented thinking.

**Table 3 behavsci-14-00692-t003:** Correlations of the BDI-II, STAI-T, TMT-B, and MWT-B with overall number of correct responses (hits) and overall response latencies for the unmasked and masked expression conditions in the emotion recognition task (Spearman rank) (*N* = 102).

Variable	BDI-II	STAI-T	TMT-B	MWT-B IQ
Unmasked hits	0.10	0.15	−0.05	0.11
Masked hits	0.04	−0.03	−0.01	0.00
Unmasked RT	0.22 *	0.09	0.19	−0.04
Masked RT	0.01	0.00	0.02	−0.07

* *p* < 0.05 (two-tailed). BDI-II = Beck Depression Inventory; STAI-T = State-Trait Anxiety Inventory—trait version; TMT-B = Trail Making Test Part B; MWT-B = Multiple-Choice Vocabulary Test Version B, intelligence quotient.

**Table 4 behavsci-14-00692-t004:** Comparison between participants starting with unmasked faces (n = 52, 34 women) and participants starting with masked faces (n = 50, 36 women) concerning alexithymia, affectivity, psychomotor functioning, and intelligence (Mann–Whitney U tests) with descriptive statistics.

Variable	Start with Unmasked Faces Mean (SD)	Start with Masked Faces Mean (SD)	Z	*p*
TAS-20 (sum score)	40.60 (10.44)	42.40 (10.89)	−1.09	0.27
TAS-20 DIF	13.62 (4.49)	14.48 (4.26)	−1.14	0.25
TAS-20 DDF	11.63 (4.29)	12.02 (4.91)	−0.32	0.75
TAS-20 EOT	15.35 (4.27)	15.9 (4.40)	−0.63	0.53
BDI-II	9.15 (4.96)	8.30 (5.30)	−0.96	0.34
STAI-T	38.00 (8.96)	38.60 (8.62)	−0.66	0.51
TMT-B	61.54 (23.38)	52.02 (14.64)	−2.00	0.046 *
MWT-B IQ	108.13 (11.55)	110.46 (11.44)	−1.31	0.19

* *p* < 0.05 (two-tailed). TAS-20 = 20-Item Toronto Alexithymia Scale; DIF = Difficulties identifying feelings; DDF = Difficulties describing feelings; EOT = Externally oriented thinking; BDI-II = Beck Depression Inventory; STAI-T = State-Trait Anxiety Inventory—trait version; TMT-B = Trail Making Test Part B; MWT-B = Multiple-Choice Vocabulary Test Version B, intelligence quotient.

**Table 5 behavsci-14-00692-t005:** Correlations of the TAS-20 scales with overall number of correct responses (hits) and overall response latencies for the unmasked and masked expression conditions in participants starting with unmasked faces (n = 52) and participants starting with masked faces in the emotion recognition task (n = 50) (Spearman rank).

Variable	TAS-20 Sum Score	TAS-20 DIF	TAS-20 DDF	TAS-20 EOT
**Start with unmasked faces**				
Unmasked hits	−0.07	−0.07	−0.03	−0.16
Masked hits	0.10	0.14	−0.05	0.17
Unmasked RT	−0.05	−0.01	0.08	−0.18
Masked RT	−0.06	−0.05	0.15	−0.23
**Start with masked faces**				
Unmasked hits	0.15	0.00	0.07	0.20
Masked hits	−0.11	−0.07	−0.21	0.01
Unmasked RT	0.26	0.18	0.22	0.24
Masked RT	0.33	0.08	0.27	0.41*

* *p* < 0.00208 (adjusted p-level, one-tailed testing). TAS-20 = 20-Item Toronto Alexithymia Scale; DIF = Difficulties identifying feelings; DDF = Difficulties describing feelings; EOT = Externally oriented thinking.

**Table 6 behavsci-14-00692-t006:** Correlations of the TAS-20 scales with response latencies for the masked emotional expression conditions (Spearman rank) in participants starting with the masked faces in the emotion recognition task (n = 50) with descriptive statistics.

Variable	TAS-20 Sum Score	TAS-20 DIF	TAS-20 DDF	TAS-20 EOT	Mean	SD
Masked Happy RT	0.10	0.13	−0.01	0.11	2.10	0.37
Masked Neutral RT	0.25	0.09	0.25	0.23	1.69	0.41
Masked Sad RT	0.25	0.14	0.16	030	2.44	0.63
Masked Disgusted RT	0.38	0.12	30	0.50 *	2.70	0.75
Masked Fearful RT	0.14	−0.08	0.15	0.18	2.40	0.65

* *p* < 0.00208 (adjusted p-level, one-tailed testing). TAS-20 = 20-Item Toronto Alexithymia Scale; DIF = Difficulties identifying feelings; DDF = Difficulties describing feelings; EOT = Externally oriented thinking.

## Data Availability

The datasets used and analyzed during the current study are available from the corresponding author on reasonable request.

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
