# Peer review of "Dimensions of Alexithymia and Identification of Emotions in Masked and Unmasked Faces"

_behavsci, 2024, doi:10.3390/bs14080692_

Round 1

Reviewer 1 Report

Comments and Suggestions for Authors

Summary. The study explored the impact of alexithymia on recognizing emotional expressions in faces covered by a mouth-nose mask. The study replicated previous findings of impaired emotion recognition in masked faces and uncovered some interesting (although moderate) effects of alexithymia on emotion recognition in masked faces.

Evaluation. In general, the methods of the study are sound and the manuscript is comprehensively written. I have one major reservation regarding the acceptance of the manuscript for publication.

There is a previous study from the same lab (essentially by the same authors) in the very same Journal where this manuscript is submitted to (Fuchs, Kersting, Suslow & Bodenschatz, 2024). However, the authors do not appropriately refer to it. They only reference it once and essentially treat it like any other study (from another work group / lab). In my opinion, this is bad practice. The authors must describe their previous study in much more detail. They must tell what they learned from the previous study and what is new in the current one. They must inform the readers of the similarities of the studies (e.g. the same stimuli, same behavioral tests). Were the samples independent or did they overlap?

Verdict. The study might be suitable for publication after a major revision.

Reviewer 2 Report

Comments and Suggestions for Authors

MAJOR COMMENTS:

After carefully reading the manuscript, I found it well-written, well-documented, and based on the provided information, it appears easy to replicate. However, my major concern is the lack of a clear hypothesis guiding the research. This issue leads to the exploration of multiple hypotheses, which increases the likelihood of finding “significant” correlations by chance. Due to the numerous correlations examined (e.g., “masked/no mask” X “first/second/both blocks” X “accuracy or latency” X “different alexithymia facets”), the probability of obtaining a false positive is very high.

The term "hypothesis" is used only once in the manuscript, specifically in the discussion section, where it reads: “Consistent with our hypothesis, we found externally oriented thinking to be linked to high response latencies for correct identifications of emotions in masked faces." Since no clear rationale was provided for this hypothesis (the introduction merely states that “the objective of the present study was to explore…”), it seems like an ad-hoc hypothesis. Furthermore, previous studies (e.g., Verroca et al.; Bagby and Taylor) did not find a correlation between alexithymia dimensions and the speed of emotion identification in faces with masks. While this discrepancy is not inherently problematic because replications in science are encouraged, given the concerns about multiple comparisons, it raises another red flag. In summary, the authors did not address the issue of multiple comparisons, which is very concerning.

Additionally, this study was conducted with healthy and "normal" participants who were screened post-hoc for different alexithymia facets. The challenge here is that most neurotypical participants selected at random, when screened for a neuropsychological phenomenon that usually co-occurs with mental disorders (such as Autism Spectrum Disorder), will likely fall within the middle of the distribution. This results in a lack of participants from the extremes of the distribution, which is necessary to provide the resolution needed to answer the research questions effectively.

MINOR COMMENTS:

1. I am curious why the authors selected different models for both conditions (with and without masks). Using the same faces (the original + the one with the engineered mask) would have ensured that the intensity (arousal) of the expressed emotion was controlled, facilitating a more accurate comparison between both conditions.

2. I am also wondering whether all the stimuli used by the authors were based on young-looking models or whether there was a mix of older and younger models. It has been found (see references listed below) that it is harder to identify the emotion expressed by older faces compared to younger faces. A disbalance between the masked and unmasked faces concerning young and old faces might therefore be concerning.

Conclusion: Altogether, I find the evidence presented in the abstract for the claim that “specific facets of alexithymia could be linked to a less efficient identification of emotions from faces wearing masks” to be very weak. While some limitations are acknowledged in the “discussion” section, the authors need to address these concerns more thoroughly to convince me that the study is worth publishing.

References:

- Calder, A. J., Keane, J., Manly, T., Sprengelmeyer, R., Scott, S., Nimmo-Smith, I., et al. (2003). Facial expression recognition across the adult life span. Neuropsychologia, 41, 195–202. doi: 10.1016/s0028-3932(02)00149-5

- Grainger, S. A., Henry, J. D., Phillips, L. H., Vanman, E. J., & Allen, R. (2017). Age deficits in facial affect recognition: the influence of dynamic cues. J. Gerontol. B Psychol. Sci. Soc. Sci., 72, 622–632.

- Grondhuis, S. N., et al. (2021). Having difficulties reading the facial expression of older individuals? Blame it on the facial muscles, not the wrinkles. Frontiers in Psychology, 12, 620768.

Round 2

Reviewer 1 Report

Comments and Suggestions for Authors

The authors made a good job in revising their manuscript. I have no further reservations in suggesting acceptance for publication. I only spotted a inconsistency in the reported p-values.

The p-value reported in line 324 does not make sense. If no significant correlations were found then all ps should be greater than 0.05 (and not greater than 0.00208). This specific numerical value appears another two-times (in the notes of Table 5 AND Table 6). Please check whether this is correct.

Author Response

Comments 1: The authors made a good job in revising their manuscript. I have no further reservations in suggesting acceptance for publication. I only spotted a inconsistency in the reported p-values.

The p-value reported in line 324 does not make sense. If no significant correlations were found then all ps should be greater than 0.05 (and not greater than 0.00208). This specific numerical value appears another two-times (in the notes of Table 5 AND Table 6). Please check whether this is correct.

Response 1: The reported p-value of 0.00208 is correct in these three cases (l. 324, in Table 5 and Table 6). Responding to a criticism made by Reviewer 2 we used a more conservative p-level in our main correlation analyses concerning the associations between alexithymia facets and emotion recognition. In section 2.5. Statistical analyses we describe our adjustment for multiple testing in detail (see l.248-255): “To adjust for multiple testing in our main correlation analyses concerning the associations of alexithymia facets with emotion recognition parameters, we used a conservative significance threshold of 0.05/24 = 0.00208 (i.e., dividing the conventional p-level by the product of the two masking conditions (masked, unmasked), the two block sequences (start with masked, start with unmasked), the two recognition parameters (accuracy, latency), and the three alexithymia facets (DIF, DDF, and EOT)). In these correlation analyses, one-tailed testing was applied because the hypotheses had a direction.“

In order to avoid misunderstandings, we reformulated the sentence in question in our Result section “Correlations of Alexithymia with Recognition Performance”, which reads now (see l.321-324):

“In the total sample, when administering the adjusted p-level of 0.00208 we found no significant correlations of the TAS-20 sum score and the scales DIF, DDF, and EOT with the overall number of correct responses and the overall response latencies for the masked and unmasked expression conditions.”

Moreover, we clarify in the footnotes of Table 5 and Table 6 that the adjusted p-level of 0.00208 was used in these correlation analyses (see l.397 and l.412).

Many thanks again to the reviewer for the efforts and time in the evaluation of our manuscript.

Reviewer 2 Report

Comments and Suggestions for Authors

Thank you for addressing my concerns. I believe the manuscript is now stronger and meets the standards for publication in Behavioral Sciences.

Author Response

Comments 1: Thank you for addressing my concerns. I believe the manuscript is now stronger and meets the standards for publication in Behavioral Sciences.

Response 1: We are delighted that we succeeded in addressing the concerns of the reviewer. Many thanks again for the valuable comments.